# ON THE THEORETICAL LIMITATIONS OF EMBEDDING-BASED RETRIEVAL

**Orion Weller**[1,2]    **Michael Boratko**[1]    **Iftekhar Naim**[1]    **Jinhyuk Lee**[1]

[1]Google DeepMind, [2]Johns Hopkins University

oweller@cs.jhu.edu,jinhyuklee@google.com

## ABSTRACT

Vector embeddings have been tasked with an ever-increasing set of retrieval tasks over the years, with a nascent rise in using them for reasoning, instruction-following, coding, and more. These new benchmarks push embeddings to work for *any query* and *any notion of relevance* that could be given. While prior works have pointed out theoretical limitations of vector embeddings, there is a common assumption that these difficulties are exclusively due to unrealistic queries, and those that are not can be overcome with better training data and larger models. In this work, we demonstrate that we may encounter these theoretical limitations in realistic settings with extremely simple queries. We connect known results in learning theory, showing that the number of top-$k$ subsets of documents capable of being returned as the result of some query is limited by the dimension of the embedding. We empirically show that this holds true even if we directly optimize on the test set with free parameterized embeddings. We then create a realistic dataset called LIMIT that stress tests embedding models based on these theoretical results, and observe that even state-of-the-art models fail on this dataset despite the simple nature of the task. Our work shows the limits of embedding models under the existing single vector paradigm and calls for future research to develop new techniques that can resolve this fundamental limitation.

## 1 INTRODUCTION

Over the last two decades, information retrieval (IR) has moved from models dominated by sparse techniques (such as BM25 Robertson et al. (1995)) to those that use neural language models (LM) as their backbones (Lee et al., 2019; Craswell et al., 2020; Izacard et al., 2021; Wang et al., 2022). These neural models are predominantly used in a single vector capacity, where they output a single *embedding* representing the entire input (also known as *dense retrieval*). These embedding models are capable of generalizing to new retrieval datasets and have been tasked with solving increasingly complicated retrieval problems (Thakur et al., 2021; Enevoldsen et al., 2025; Lee et al., 2025).

In recent years this has been pushed even further with the rise of instruction-following retrieval benchmarks, where models are asked to represent **any relevance definition** for **any query** (Weller et al., 2025a;c; Song et al., 2025; Xiao et al., 2024; Su et al., 2024). For example, the QUEST dataset (Malaviya et al., 2023) uses logical operators to combine different concepts, studying the difficulty of retrieval for complex queries (e.g., "Moths or Insects or Arthropods of Guadeloupe"). On the other hand, datasets such as BRIGHT (Su et al., 2024) explore the challenges arising from different definitions of relevance by defining relevance in ways that require reasoning. One subtask includes reasoning over a given Leetcode problem (the query) to find other Leetcode problems that share a subtask (e.g. others problems using dynamic programming). Although models cannot solve these benchmarks yet, the community has proposed these problems in order to push the boundaries of what dense retrievers are capable of—which is now implicitly *every task* that could be defined.

Rather than proposing empirical benchmarks to gauge what embedding models can achieve, we seek to understand at a more fundamental level what the limitations are. Since embedding models use

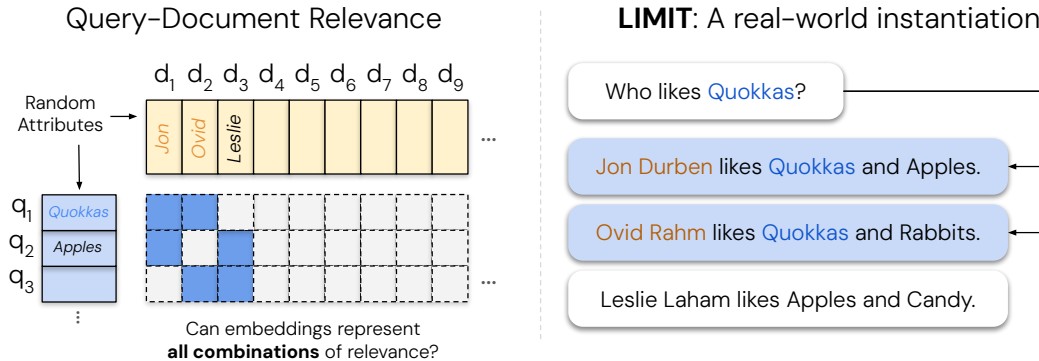

Figure 1: A depiction of the LIMIT dataset creation process, based on theoretical limitations. We test **all combinations** of relevance for $N$ documents (i.e. in the figure, all combinations of relevance for three documents with two relevant documents per query) and instantiate it using a simple mapping.

vector representations in geometric space, there exist well-studied fields of mathematical research (Papadimitriou & Sipser, 1982) that could be used to analyze these representations.

Our work aims to bridge this gap, connecting known theoretical results in linear algebra with modern advancements in neural information retrieval. We draw upon research in high-dimensional geometry to provide a lower bound on the embedding dimension needed to represent a given combination of relevant documents and queries. Specifically, we show that for a given embedding dimension $d$ **there exist top-$k$ combinations of documents that cannot be returned**—no matter the query—highlighting a theoretical and fundamental limit to embedding models.

To show that this theoretical limit is true for any retrieval model or training dataset, we test a setting where the vectors themselves are directly optimized with the test data. This allows us to empirically show how the embedding dimension enables the solving of retrieval tasks. We find that there exists a crucial point for each embedding dimension ($d$) where the number of documents is too large for the embedding dimension to encode all combinations. We then gather these crucial points for a variety of $d$ and show that this relationship can be modeled empirically with a polynomial function.

We also go one step further and construct a realistic but simple dataset based on these theoretical limitations (called LIMIT).[1] Despite the simplicity of the task (e.g., `who likes Apples?` and `Jon likes Apples, ...`), we find it is very difficult for even state-of-the-art embedding models (Lee et al., 2025; Zhang et al., 2025) on MTEB (Enevoldsen et al., 2025), and practically impossible for models with small embedding dimensions using standard optimization techniques.

Overall, our work contributes: (1) a theoretical basis for the fundamental limitations of embedding models, (2) a best-case empirical analysis showing that this proof holds for any dataset instantiation (by free embedding optimization), and (3) a simple real-world natural language instantiation called LIMIT that even state-of-the-art embedding models cannot solve.

These results imply interesting findings for the community: on one hand we see neural embedding models becoming immensely successful. However, academic benchmarks test only a small amount of the queries that could be issued (and these queries are often overfitted to), hiding these limitations. Our work shows that as the tasks given to embedding models require returning ever-increasing combinations of top-$k$ relevant documents (e.g., through instructions connecting previously unrelated documents with logical operators), we will reach a limit of combinations they can represent.

Thus, the community should be aware of these limitations, both when creating evals and also by using alternate architectures—such as cross-encoders / multi-vector / more expressive similarity functions —when trying to handle the full range of instruction queries, i.e. *any query and relevance definition*.

---

[1]Data and code are available at `https://github.com/google-deepmind/limit`

## 2 RELATED WORK

### 2.1 NEURAL EMBEDDING MODELS

There has been immense progress on embedding models in recent years (Lee et al., 2019; Craswell et al., 2020; BehnamGhader et al., 2024), moving from simple web search (text-only) to advanced instruction-following and multi-modal representations. These models generally followed advancements in language models, such as pre-trained LMs (Hoffmann et al., 2022), multi-modal LMs (Li et al., 2024; Team, 2024), and advancements in instruction-following (Zhou et al., 2023; Ouyang et al., 2022). Some of the prominent examples in retrieval include CoPali (Faysse et al., 2024) and DSE (Ma et al., 2024) which focus on multimodal embeddings, Instructor (Su et al., 2022) and FollowIR (Weller et al., 2024a) for instruction following, and GritLM (Muennighoff et al., 2024) and Gemini Embeddings (Lee et al., 2025) for pre-trained LMs turned embedders.

Our work, though focused solely on textual representations for simplicity, **applies to all modalities of single vector embeddings for any domain of dataset**. As the space of things to represent grows (through instructions or multi-modality) they will increasingly run into these theoretical limitations.

### 2.2 EMPIRICAL TASKS PUSHING THE LIMITS OF DENSE RETRIEVAL

Retrieval models have been pushed beyond their initial use cases to handle a broad variety of areas. Notable works include efforts to represent a wide group of domains (Thakur et al., 2021; Lee et al., 2024), a diverse set of instructions (Weller et al., 2024a; Zhou et al., 2024; Oh et al., 2024; Weller et al., 2025b), and to handle reasoning over the queries (Xiao et al., 2024; Su et al., 2024). This has pushed the focus of embedding models from basic keyword matching to embeddings that can represent the full semantic meaning of language. As such, it is more common than ever to connect what were previously unrelated documents into the top-$k$ relevant set,[2] increasing the number of combinations that models must be able to represent. This has motivated our interest in understanding the limits of what embeddings can represent, as current work expects it to handle *every* task.

Previous work has explored empirically the limits of models: Reimers & Gurevych (2020) showed that smaller dimension embedding models have more false positives, especially with larger-scale corpora. Ormazabal et al. (2019) showed the empirical limitations of models in the cross-lingual setting and Yin & Shen (2018) showed how embedding dimensions relate to the bias-variance tradeoff. In contrast, our work provides a theoretical connection between the embedding dimension and the top-k sets it can retrieve, while also showing empirical limitations.

### 2.3 THEORETICAL LIMITS OF VECTORS IN GEOMETRIC SPACE

Understanding and finding nearest neighbors in semantic space has a long history in mathematics research, with early work such as the Voronoi diagram being studied as far back as 1644 and formalized in 1908 (Voronoi, 1908). The order-$k$ version of the Voronoi diagram (i.e. the Voronoi diagram partitioning the space into regions based on their closest $k$ points) is obviously connected to information retrieval and has been studied for many years (Clarkson, 1988). The number of such regions is equal to the number of unique retrieval sets of size $k$, however this quantity is notoriously difficult to bound tightly (Bohler et al., 2015; Lee, 1982; Chen et al., 2023).

We approach this problem from a different angle, asking not how many $k$-subsets a given configuration realizes, but rather what embedding dimension is *necessary* to realize all $k$-subsets with a guaranteed score margin. By applying a classical sphere-packing volume argument (Vershynin, 2018; Conway et al., 1999), we obtain a lower bound on the embedding dimension in terms of $n$, $k$, and the margin $\gamma$. Our result is conceptually related to the Johnson–Lindenstrauss lemma (Johnson et al., 1984), which gives a *sufficient* dimension to preserve pairwise distances among $n$ points; in contrast, our bound gives a *necessary* dimension to realize all retrieval sets with a margin. The role of the margin in controlling the complexity of realizable configurations also parallels classical results in statistical learning theory, including the fat-shattering dimension (Kearns & Schapire, 1994) and margin-based generalization bounds for linear classifiers (Bartlett, 2002; Vapnik, 1998), where larger margins similarly constrain the capacity of the hypothesis class.

---

[2]You can imagine an easy way to connect any two documents merely by using logical operators, i.e. X and Y.

## 3 Representational Capacity of Vector Embeddings

In this section we formally define the minimum embedding dimension required to satisfy a given retrieval objective, and draw on classical sphere-packing results from high-dimensional geometry to establish a lower bound. We note that this will be an extreme lower bound, as practical models have to deal with other constraints such as learning through gradient descent and using LM tokenization.

**Setup.** Let $v_1, \ldots, v_n \in \mathbb{R}^d$ be unit[3] document vectors, and let queries be unit vectors $u \in \mathbb{R}^d$. Fix $\gamma > 0$. A $k$-subset $S \subseteq [n]$ is realized with margin $\gamma$ if there exists a unit query $u_S$ such that

$$\min_{i \in S} \langle u_S, v_i \rangle \geq \max_{j \notin S} \langle u_S, v_j \rangle + 2\gamma. \tag{1}$$

Since $\langle u, v_i \rangle \in [-1, 1]$ for unit vectors, any score gap is at most 2, hence equation 1 is feasible only for $0 < \gamma \leq 1$. Throughout, $\log$ denotes the natural logarithm.

**Theorem 1** (Dimension lower bound). *Assume $1 \leq k < n$ and that* every *$k$-subset $S \subseteq [n]$ is realized with margin $\gamma$ as in equation 1. Then*

$$\binom{n}{k} \leq \left(1 + \frac{1}{\gamma}\right)^d, \qquad \text{hence} \qquad d \geq \frac{\log \binom{n}{k}}{\log(1 + 1/\gamma)}. \tag{2}$$

*Proof.* Fix two distinct $k$-subsets $S \neq T$ and choose $i \in S \setminus T$ and $j \in T \setminus S$. Applying equation 1 to $S$ and to $T$ gives

$$\langle u_S, v_i - v_j \rangle \geq 2\gamma, \qquad \langle u_T, v_j - v_i \rangle \geq 2\gamma.$$

Adding yields $\langle u_S - u_T, v_i - v_j \rangle \geq 4\gamma$. By Cauchy–Schwarz and (universally, for *any* unit vectors) $\|v_i - v_j\| \leq \|v_i\| + \|v_j\| = 2$, we obtain $\|u_S - u_T\| \geq 2\gamma$. Thus the $M = \binom{n}{k}$ unit queries $\{u_S\}$ are pairwise $2\gamma$-separated, so the open balls $B(u_S, \gamma)$ are disjoint. Moreover, since $\|u_S\| = 1$, each $B(u_S, \gamma) \subseteq B(0, 1 + \gamma)$, and therefore

$$M \cdot \mathrm{vol}\big(B_d(\gamma)\big) \leq \mathrm{vol}\big(B_d(1 + \gamma)\big).$$

Using $\mathrm{vol}(B_d(r)) = C_d\, r^d$ for a constant $C_d$ depending only on $d$ (which cancels), we get $M\gamma^d \leq (1 + \gamma)^d$, i.e. $\binom{n}{k} \leq \big((1 + \gamma)/\gamma\big)^d = \big(1 + 1/\gamma\big)^d$. Rearranging yields equation 2. $\square$

| Corpus size $n$ | $k = 2$ | $k = 10$ | $k = 100$ | $k = 1000$ |
|---|---|---|---|---|
| $10^2$ | 4 | 13 | *trivial* | — |
| $10^3$ | 6 | 23 | 135 | *trivial* |
| $10^4$ | 8 | 33 | 233 | 1354 |
| $10^5$ | 10 | 42 | 329 | 2334 |
| $10^6$ | 12 | 52 | 425 | 3296 |
| $10^7$ | 14 | 61 | 521 | 4257 |
| $10^8$ | 16 | 71 | 617 | 5217 |
| $10^9$ | 17 | 81 | 713 | 6177 |
| $10^{10}$ | 19 | 90 | 809 | 7137 |
| $10^{11}$ | 21 | 100 | 905 | 8098 |

Table 1: Lower bounds for embedding dimension from Theorem 1 for $\gamma = 0.1$. When $n$ and $k$ are both 1000, the result is *trivial* because $\binom{1000}{1000} = 1$ (there is only one $k$-subset, hence no "irrelevant" items to separate). We see for large $k$ and $n$ values these dimension requirements are already greater than those currently used for web-scale search. **If these numbers are inflated by a small multiple due to constraints on gradient learning or other LM-based constraints** (e.g. tokenization, generalization) **these bounds are outside of any reasonable embedding dimension.**

---

[3]For simplicity, as nearly all SoTA retrieval models use unit vectors.

### 3.0.1 IMPLICATIONS

**Numerical instantiation**  We can illustrate the effects of this lower bound using $\gamma = 0.1$ (score gap $2\gamma = 0.2$), which is approximately standard for models based on empirical usage. Thus, equation 2 becomes $d \geq \lceil \log \binom{n}{k} / \log 11 \rceil$, with a table for various $k$ and $n$ values in Table 1.

For $n \gg k$, $\log \binom{n}{k} \approx k \log(en/k)$, so equation 2 forces

$$d \;=\; \Omega\!\left(\frac{k \log(en/k)}{\log(1 + 1/\gamma)}\right).$$

A stricter margin requirement (larger $\gamma$) demands higher dimension, since $\log(1 + 1/\gamma)$ decreases with $\gamma$ (feasibility requires $\gamma \leq 1$, so the denominator is at least $\log 2$).

**Consequences**  Due to space and speed requirements, most embeddings used for web-scale search are quantized or truncated (e.g. through Matryoshka embeddings (Kusupati et al., 2022)) to less than 1k dimensions, while the largest embeddings used in research are around 4096 (Zhang et al., 2025). We see that even with a moderate margin, which is needed to handle noise from messy data or quantization, the lower bounds in Table 1 can already be larger than what is used in practice.

Additional constraints on real-world models (such as needing to generalize, learn from gradient descent, and use natural language and tokenization) will make the dimension required in practice much higher. As Table 1 shows, even a small multiple of this lower bound would make the embedding dimension requirement infeasible. This multiple seems well-founded, as we will show in the next section from the best-case optimization setting (e.g. free embeddings).

## 4 EMPIRICAL CONNECTION: BEST CASE OPTIMIZATION

Having established a theoretical limitation of embedding models based on their embedding dimension $d$, we seek to show that this holds empirically also.

To show the strongest optimization case possible, we design experiments where the vectors themselves are directly optimizable with gradient descent.[4] We call this "free embedding" optimization, as the embeddings are free to be optimized and not constrained by natural language, which imposes constraints on any realistic embedding model. Thus, this shows whether it is feasible for **any embedding model** to solve this problem: if the free embedding optimization cannot solve the problem, real retrieval models will not be able to either. It is also worth noting that we do this by directly optimizing the embeddings over the target qrel matrix (test set). This will not generalize to a new dataset, but is done to show the highest performance that could possibly occur.

**Experimental Settings**  We create a random document matrix (size $n$) and a random query matrix with top-$k$ sets (of all combinations, i.e. size $m = \binom{n}{k}$), both with unit vectors. We then directly optimize for solving the constraints with the Adam optimizer (Kingma & Ba, 2014).[5] Each gradient update is a full pass through all correct triples (i.e. full dataset batch-size) with the InfoNCE loss function (Oord et al., 2018),[6] with all other documents as in-batch negatives (i.e. full dataset in batch). As nearly all embedding models use normalized vectors, we do also (via projected gradient descent). We perform early stopping when there is no improvement in the loss for 1000 iterations. We gradually increase the number of documents (and thus the binomial amount of queries) until the optimization is no longer able to solve the problem (i.e. achieve 100% accuracy). We call this the *critical-n* point.

We focus on relatively small sizes for $n$, $k$, and $d$ due to the combinatorial explosion of combinations with larger document values (i.e. 50k docs with top-$k$ of 100 gives 7.7e+311 combinations, which would be equivalent to the number of query vectors of dimension $d$ in that free embedding experiment).

---

[4]This could also be viewed as an embedding model where each query/doc are a separate vector via lookup.

[5]We found similar results with SGD, but we use Adam for speed and similarity with existing training methods.

[6]In preliminary experiments, we found that InfoNCE performed best, beating MSE and Margin. As we are directly optimizing the vectors with full-dataset batches, this is $\mathcal{L}_{\text{total}} = -\frac{1}{M} \sum_{i=1}^{M} \log \frac{\sum_{d_r \in R_i} \exp(\text{sim}(q_i, d_r)/\tau)}{\sum_{d_k \in D} \exp(\text{sim}(q_i, d_k)/\tau)}$ where $D$ is all docs, $d_r$ is the relevant documents for query $q_i$ and $d_k$ are the non-relevant documents. For experiments with sigmoid learning functions (e.g. Bangachev et al. (2025), see Appendix C)

We use $k = 2$ and increase $n$ by one for each $d$ value until it breaks. We fit a polynomial regression line to the data so we can model and extrapolate results outwards.

**Results** Figure 2 shows that the curve fits a 3rd degree polynomial curve, with formula $y = -10.5322 + 4.0309d + 0.0520d^2 + 0.0037d^3$ ($r^2$=0.999). Extrapolating this curve outward gives the critical-n values (for embedding size): 500k (512), 1.7m (768), 4m (1024), 107m (3072), 250m (4096). We note that this is the best case: a real embedding model cannot directly optimize the query and document vectors to match the test qrel matrix (and is constrained by factors such as "modeling natural language"). The results also show that the lower bounds in the previous section are a gross underestimate of real-world performance, as Table 1 shows a lower bound of 4 for $n = 100$ whereas we see the free embeddings needing $d > 18$ (e.g. a 4.5 multiplier even in the no-generalization or natural language case). Overall, these numbers already show that for web-scale search, even the largest embedding dimensions with ideal test-set optimization are not enough to model all combinations.

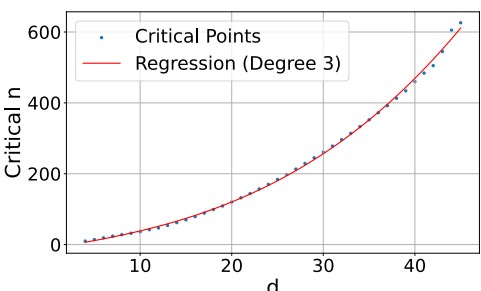

Figure 2: The critical-n value where the dimensionality is too small to successfully represent all the top-2 combinations. We plot the trend line as a polynomial function.

## 5 EMPIRICAL CONNECTION: REAL-WORLD DATASETS

The free embedding experiments provide empirical evidence that our theoretical results hold true. However, they still are abstract - what does this mean for real embedding models? In this section we (1) draw connections from this theory to existing datasets and (2) create a trivially simple yet extremely difficult retrieval task for existing SOTA models.

### 5.1 CONNECTION TO EXISTING DATASETS

Existing retrieval datasets typically use a static evaluation set with limited numbers of queries, as relevance annotation is expensive to do for each query. This means practically that the space of queries used for evaluation is a very small sample of the number of potential queries. For example, the QUEST dataset (Malaviya et al., 2023) has 325k documents and queries with 20 relevant documents per query, with a total of 3357 queries. The number of unique top-20 document sets that could be returned with the QUEST corpus would be $\binom{325k}{20}$ which is equal to 7.1e+91 (larger than the estimate of atoms in the observable universe, $10^{82}$). Thus, the 3k queries in QUEST can only cover an infinitesimally small part of the qrel combination space.

Although it is not possible to instantiate all combinations when using large-scale corpora, search evaluation datasets are a proxy for what any user would ask for and ideally would be designed to test many combinations, as users will do. In many cases, developers of new evaluations simply choose to use fewer queries due to cost or computational expense of evaluation. For example, QUEST's query "Novels from 1849 or George Sand novels" combines two categories of novels with the "OR" operator – one could instantiate new queries to relate concepts through OR'ing other categories together. Similarly, with the rise of search agents, we see greater usage of hyper-specific queries: BrowseComp (Wei et al., 2025) has 5+ conditions per query, including range operators. With these tools, it is possible to sub-select any top-$k$ relevant set with the right operators if the documents are sufficiently expressive (i.e. non-trivial). Thus, that existing datasets choose to only instantiate some of these combinations is mainly for practical reasons and not because of a lack of existence.

In contrast to these previous works, we seek to build a dataset that evaluates all combinations of top-$k$ sets for a small number of documents. Rather than using difficult query operators like QUEST, BrowseComp, etc. (which are already difficult for reasons outside of the qrel matrix) we choose very simple queries and documents to highlight the difficulty of representing all top-$k$ sets themselves.

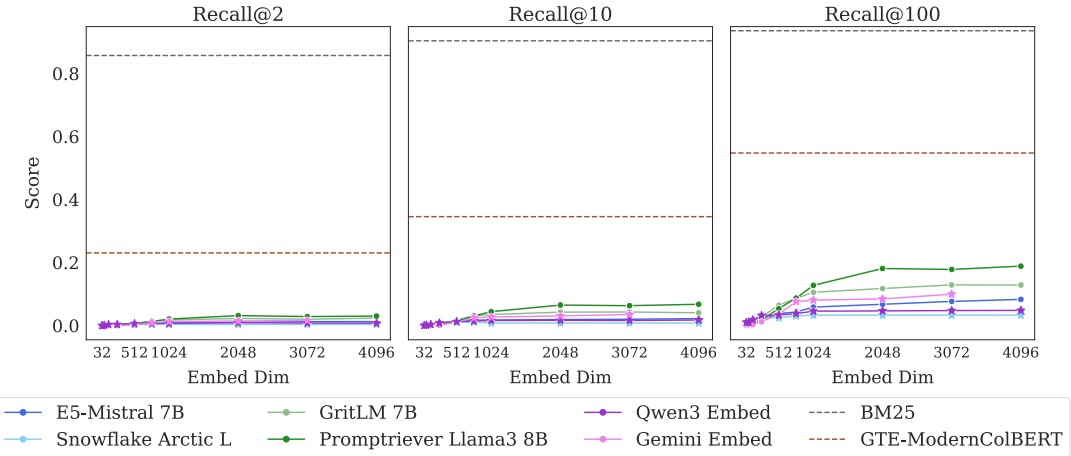

Figure 3: Scores on the LIMIT task. Despite the simplicity of the task we see that SOTA models struggle. We also see that the dimensionality of the model is a limiting factor and that as the dimension increases, so does performance. Even multi-vector models struggle. Lexical models like BM25 do very well due to their higher dimensionality. Stars indicate models trained with MRL.

## 5.2 THE LIMIT DATASET

**Dataset Construction** In order to have a natural language version of this dataset, we need some way to map combinations of documents into something that could be retrieved with a query. One simple[7] way to do this is to create a synthetic version with latent variables for queries and documents and then instantiate it with natural language. For this mapping, we choose to use attributes that someone could like (i.e. Jon likes Hawaiian pizza, sports cars, etc. ) as they are plentiful and don't present issues w.r.t. other items: one can like Hawaiian pizza but dislike pepperoni, all preferences are valid. We then enforce two constraints for realism: (1) users shouldn't have too many attributes, thus keeping the documents short (less than 50 per user) and (2) each query should only ask for one item to keep the task simple (i.e. "who likes X"). We gather a list of attributes a person could like through prompting Gemini 2.5 Pro. We then clean it to a final 1850 items by iteratively asking it to remove duplicates/hypernyms, while also checking the top failures with BM25 to ensure no overlap.

We choose to use 50k documents in order to have a hard but relatively small corpus and 1000 queries to maintain statistical significance while still being fast to evaluate. For each query, we choose to use two relevant documents (i.e. $k$=2), both for simplicity in instantiating and to mirror previous work (i.e. NQ, HotpotQA, etc. (Kwiatkowski et al., 2019; Yang et al., 2018)).

Our last step is to choose a qrel matrix to instantiate these attributes. Although we could not prove the hardest qrel matrix definitively with theory, we intuit that our theoretical results imply that the more interconnected the qrel matrix is (e.g. dense with all combinations) the harder it would be for models to represent. Following this, we use the qrel matrix with the highest number of documents for which all combinations would be just above 1000 queries for a top-$k$ of 2 (46 docs, since $\binom{46}{2}$ is 1035, the smallest above 1k).

We then assign random natural language attributes to the queries, adding these attributes to their respective relevant documents (c.f. Figure 1). We give each document a random first and last name from open-source lists of names. Finally, we randomly sample new attributes for each document until all documents have the same number of attributes. As this setup has many more documents than those that are relevant to any query (46 relevant documents, 49.95k non-relevant to any query) we also create a "small" version with only the 46 documents that are relevant to one of the 1000 queries.

**Models** We evaluate the state-of-the-art embedding models including GritLM (Muennighoff et al., 2024), Qwen 3 Embeddings (Zhang et al., 2025), Promptriever (Weller et al., 2024b), Gemini Embeddings (Lee et al., 2025), Snowflake's Arctic Embed Large v2.0 (Yu et al., 2024), and E5-

---

[7]This is just one way, designed to be realistic and simple. However, our framework allows for any way of instantiation – not stuck to this arbitrary natural language design.

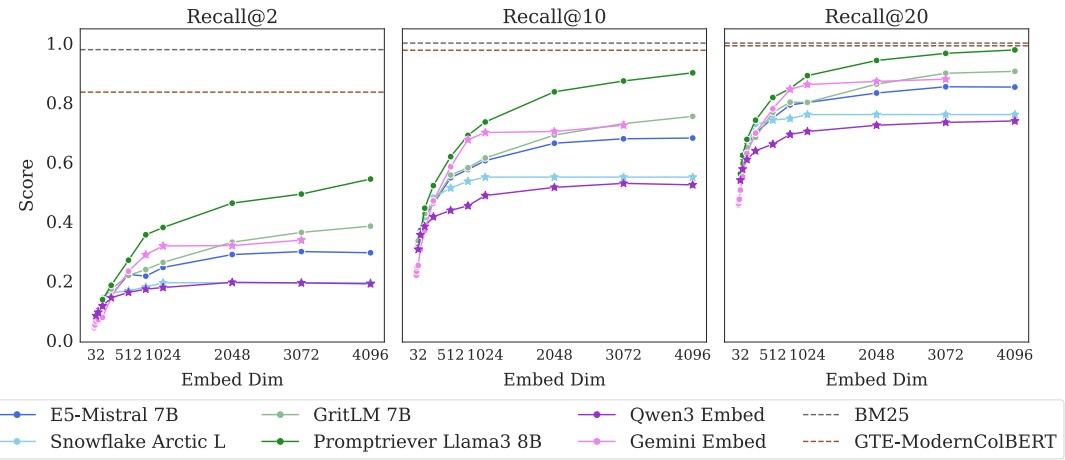

Figure 4: Scores on the LIMIT small task (N=46) over embedding dimensions. Despite having just 46 documents, models struggle even with recall@10 and cannot solve the task even with recall@20.

Mistral Instruct (Wang et al., 2022; 2023). These models range in embedding dimension (1024 to 4096) as well as in training style (instruction-based, hard negative optimized, etc.). We also evaluate three non-single vector models to show the distinction: BM25 (Robertson et al., 1995; Lù, 2024), gte-ModernColBERT (Chaffin, 2025a; Chaffin & Sourty, 2024), and a token-wise TF-IDF.[8]

We show results at the full embedding dimension and also with truncated embedding dimension (typically used with matryoshka learning, aka MRL (Kusupati et al., 2022)). For models not trained with MRL this will result in sub-par scores, thus, models trained with MRL are indicated with stars in the plots. However, as there are no LLMs with an embedding dimension smaller than 384, we include MRL for all models to small dimensions (32) to show the impact of embedding dimensionality.

**Results** Figure 3 shows the results on the full LIMIT while Figure 4 shows the results on the small (46 document) version. **The results are surprising - models severely struggle even though the task is trivially simple.** For example, in the full setting models struggle to reach even 20% recall@100 and in the 46 document version models cannot solve the task even with recall@20.

We see that model performance depends crucially on the embedding dimensionality (better performance with bigger dimensions). Interestingly, models trained with more diverse instruction, such as Promptriever, perform better, perhaps because their training allows them to use more of their embedding space (compared to models which are trained with MRL and on a smaller range of tasks that can perhaps be consolidated into a smaller embedding manifold).

For alternative architectures, GTE-ModernColBERT does significantly better than single-vector models (although far from solving the task) while BM25 comes close to perfect scores. Both of these alterative architectures (sparse and multi-vector) offer various trade-offs, see §5.3 for analysis.

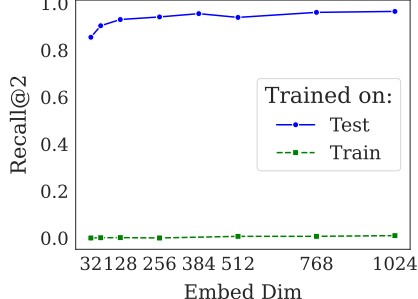

Figure 5: Training on LIMIT train does not significantly help, indicating the issue is not domain shift. But models can solve it if they overfit to the test set.

**Is this Domain Shift?** Although our queries look similar to standard web search queries, we wondered whether there could be some domain shift causing the low performance. If so, we would expect that training on a training set of similar examples would significantly improve performance. On the other hand, if the task was intrinsically hard, training on the training set would provide little help whereas training on the test set would allow the model to overfit to those tokens (similar to the free embeddings).

To test this we take an off-the-shelf embedding model and train it on either the training set (created synthetically using non-test set attributes) or the official test set of LIMIT. We use `lightonai/modernbert-embed-large` (Chaffin, 2025c) and fine-tune it on these splits, us-

---

[8]This model turns each unique item into a token and then does TF-IDF. We build it to show that it gets 100% on all tasks (as it reverse engineers our dataset construction) and thus we do not include it in future charts.

Figure 6: Comparing scores on LIMIT small vs LIMIT small (synonym). Using synonyms makes the task harder so all models perform worse. However, the lexical model (BM25) drops nearly 90%, performing worse than most single-vector models on the synonym version of LIMIT. Thus, lexical models have weaknesses of their own (see Section 5.3 for more discussion) and are not a panacea.

ing the full dataset for in batch negatives (excluding positives) using SentenceTransformers (Reimers & Gurevych, 2019). We show a range of dimensions by projecting the hidden layer down to the specified size during training (rather than using MRL).

Figure 5 shows the model trained on the training set cannot solve the problem, although it does see very minor improvement from near zero recall@10 to up to 2.8 recall@10. The lack of performance gains when training in-domain indicate that poor performance is not due to domain shift. By training the model on the test set we see it can learn the task, overfitting on the tokens in the test queries. This aligns with our free embedding results, that it is possible to overfit to the $N = 46$ version with only 12 dimensions. However, it is notable that the real models with 64 dimensions still cannot completely solve the task, implying **real models perform significantly worse than the bounds shown in §4**.

**What about Non-Lexical Matches?**    Our previous results show that lexical models greatly outperform their neural counterparts. However, this is not to imply that lexical models are a panacea - although they have higher dimensionality than single-vector models, they have other shortcomings.

We illustrate this by creating a version of LIMIT-small that replaces all items in the corpus with their synonyms, reducing the amount of lexical overlap. We ask Gemini 2.5 Pro to come up with synonyms that don't match any other existing synonyms or original items, by using either scientific names, similar meanings, or if necessary hypernyms. This creates a mapping like "glasses" to "spectacles", etc.[9] We repeat the previous experiment on LIMIT-small (synonyms) and compare to LIMIT-small in Figure 6. We find that all models drop in performance as the task is now more difficult, but BM25 drops the most and now underperforms the neural models (e.g. BM25 drops more than 89% whereas Qwen3 embedding drops 38.9%). Thus, we can see that although lexical models have strengths like higher dimensionality, they are limited by their keyword-only matching ability. We expand more upon their strengths and weaknesses for instruction-following in Section 5.3.

**Implications**  Single-vector models are fundamentally limited by their embedding dimension. The LIMIT dataset is a particular instantiation, with very simple queries and documents, designed to highlight this property. This small version of LIMIT can be embedded in just 12 dimensions (as seen in the free embeddings experiments), yet all models fail to perform well, suggesting other architectural weaknesses. Irrespective of the architecture involved, however, our framework can scale the dataset's difficulty to consistently demonstrate this fundamental limitation.

## 5.3 ALTERNATIVES TO EMBEDDING MODELS

Our previous results show both theoretically and empirically that embedding models cannot represent all combinations of documents in their top-$k$ sets, making them unable to represent and solve some retrieval tasks. As current embedding models have grown larger (e.g. up to 4096), this has helped reduce negative effects for smaller dataset sizes. However, with enough combinations of top-$k$ sets the dimensionality would have to increase to an infeasible size for non-toy datasets. Thus, although they are useful for first stage results, more expressive retriever architectures will be needed.

**Cross-Encoders**  Although not suitable for first stage retrieval at scale, they are already typically used to improve first stage results. Is LIMIT challenging for rerankers also? We evaluate a long context reranker, Gemini-2.5-Pro (Comanici et al., 2025) on the small setting as a comparison. We give Gemini all 46 documents and all 1000 queries at once, asking it to output the relevant documents for each query with one generation. We find that it can successfully solve (100%) all 1000 queries in one forward pass. This is in contrast to even the best embedding models with a recall@2 of less than

---

[9]We note that it doesn't remove all lexical overlap due to items like "Scuba Diving" -> "Underwater Diving"

60% (Figure 4). Thus we can see that LIMIT is easy for state-of-the-art reranker models, which do not have the same limitations based on embedding dimension.

**Multi-vector models** Multi-vector models are more expressive through the use of multiple vectors per sequence combined with the MaxSim operator (Khattab & Zaharia, 2020). These models show promise on the LIMIT dataset, with scores greatly above the single-vector models despite using a smaller backbone (ModernBERT, Warner et al. (2024)). However, these models are not generally used for instruction-following or reasoning-based tasks (see Chaffin (2025b) as one of the few that exist), leaving it an open question to how well multi-vector techniques will transfer to these tasks.

**Sparse models** Sparse models (both lexical and neural) can be thought of as single vectors but with very high dimensionality. This dimensionality helps BM25 avoid the problems of the neural embedding models as seen in Figure 3. Since the $d$ of their vectors is high, they can scale to many more combinations than their dense vector counterparts. However, it is less clear how to apply sparse models to instruction-following and reasoning-based tasks where there is no lexical or even paraphrase-like overlap. We leave this direction (and hybrid sparse/dense solutions) to future work.

We note that all of these options have various trade-offs and none provide a clear path to solving this problem as-is. We leave it to future work to develop new techniques to mitigate these issues: perhaps through one of these alterative categories or through new ideas around single-vector models that can resolve the underlying issue (potentially through techniques such as hyperencoders (Killingback et al., 2025) or other future work on single vector architectures yet to be developed).

## 6 CONCLUSION

We introduce the LIMIT dataset, which highlights a fundamental limitation of embedding models. We provide a theoretical connection which shows that, for a fixed embedding dimension there will be some set of documents such that certain sets are unattainable as top-$k$ sets. We show these theoretical results hold empirically, through best case optimization of the vectors themselves, and make a practical connection to existing state-of-the-art models by creating a realistic and simple instantiation of the theory, called LIMIT, that these models cannot solve. Our results imply that the community should reconsider how instruction-based retrieval will impact future retrievers.

## LIMITATIONS

Although our experiments provide theoretical insight for the most common type of embedding model (single vector) they do not hold necessarily for other architectures, such as multi-vector models. Although we showed initial empirical results with non-single vector models, we leave it to future work to extend our theoretical connections to these settings. We also did not show theoretical results for the setting where the user allows some mistakes, e.g. capturing only the majority of the combinations. We leave putting a bound on this scenario to future work and would invite the reader to examine works like Ben-David et al. (2002).

We have shown the theoretical connection that proves that some combinations cannot be represented by embedding models, however, we cannot prove apriori which *types* of combinations they will fail on. Thus, it is possible that there are some instruction-following or reasoning tasks they can solve perfectly, however, *we do know* that there exists some tasks that they will never be able to solve.

## ACKNOWLEDGMENTS

We thank Tanmaya Dabral, Zhongli Ding, Anthony Chen, Ming-Wei Chang, Kenton Lee, and Kristina Toutanova for their helpful feedback. We thank Kiril Bangachev, Guy Bresler, Iliyas Noman, Yury Polyanskiy, Antonio Vergari, Adam Lopez, and Andreas Grivas for pointers to work on sign-rank.

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
