## A    RELATIONSHIP TO ORDER-K VORONOI REGIONS

We also provide an explanation for how our results compare to Clarkson (1988) which put bounds on the number of regions in the order-$k$ Voronoi graph. The order-$k$ Voronoi graph is defined as the set of points having a particular set of $n$ points in $S$ as its $n$ nearest neighbors. This maps nicely to retrieval, as each order-$k$ region is equivalent to one retrieved set of top-$k$ results. Then the count of unique regions in the Voronoi graph is the total number of combinations that could be returned for those points. However, creating an empirical order-k Voronoi graph is computationally infeasible for $d > 3$, and theoretically it is hard to bound tightly. Thus we use a different approach for showing the limitations of embedding models.

## B    HYPERPARAMETER AND COMPUTE DETAILS

**Inference**    We use the default length settings for evaluating models using the MTEB framework (Enevoldsen et al., 2025). As our dataset has relatively short documents (around 100 tokens), this does not cause an issue.

**Training**    For training on the LIMIT training and test set we use the SentenceTransformers library (Reimers & Gurevych, 2019) using the MultipleNegativesRankingLoss. We use a full dataset batch size and employ the no duplicates sampler to ensure that no in-batch negatives are duplicates of the positive docs. We use a learning rate of 5e-5. We train for 5 epochs and limit the training set slightly to the size of the test set (from 2.5k to 2k examples, matching test).

**Compute**    Inference and training for LIMIT is done with A100 GPUs on Google Colab Pro. The free embedding experiments are done mainly on H100 GPUs and TPU v5's for larger size $N$ to accommodate higher VRAM for full-dataset batch vector optimization.

## C    SIGMOID LOSS FUNCTION FOR FREE EMBEDDINGS

In concurrent work by Bangachev et al. (2025), they show that for vision-language embedding models like CLIP (that more commonly use sigmoid loss functions) that the free-embedding experiments can be solved in fewer dimensions than in our setting (assuming no margin). As our results found the best performance with InfoNCE, which attempts to create the widest possible margin, this indicates that there are additional questions to resolve around learnability. We welcome further insight into this question both theoretically and empirically, as there exists widely disparate practices between the vision-language community (where sigmoid loss functions are often SOTA) and the text-only community (where sigmoid loss functions are almost never used due to worse performance).

This sigmoid learning is also closely related to other work, such as Grivas et al. (2024) and generally on the topic of work such as (Badreddine et al., 2025; Paul et al., 2021)

## D    PROOF USING SIGN-RANK

In the initial version of this paper, we provided a theoretical bound without any margin requirement based on the qrel matrices *sign rank*. Although the proof is correct, the sign rank of the $\binom{n}{k}$ matrix has been established in previous work (Alon et al., 1985) and only depends on k. We include the proof connecting notions relevant for retrieval with the classic notion of sign-rank, however we emphasize that this will provide a weaker requirement on dimension as it assumes no margin.

### D.1    FORMALIZATION

We consider a set of $m$ queries and $n$ documents with a ground-truth relevance matrix $A \in \{0,1\}^{m \times n}$, where $A_{ij} = 1$ if and only if document $j$ is relevant to query $i$.[10] Vector embedding models map each query to a vector $u_i \in \mathbb{R}^d$ and each document to a vector $v_j \in \mathbb{R}^d$. Relevance is modeled by the dot product $u_i^T v_j$, with the goal that relevant documents should score higher than irrelevant ones.

---

[10]The matrix $A$ is often called the "qrels" (query relevance judgments) matrix in information retrieval.

Concatenating the vectors for queries in a matrix $U \in \mathbb{R}^{d \times m}$ and those for documents in a matrix $V \in \mathbb{R}^{d \times n}$, these dot products are the entries of the score matrix $B = U^T V$. The smallest embedding dimension $d$ that can realize a given score matrix is, by definition, the rank of $B$. Therefore, our goal is equivalent to finding the minimum rank of a score matrix $B$ that correctly orders documents according to the relevance specified in $A$, which we formalize in the following definition.

**Definition 1.** Given a matrix $A \in \mathbb{R}^{m \times n}$, the **row-wise order-preserving rank of** $A$ is the smallest integer $d$ such that there exists a rank-$d$ matrix $B$ that preserves the relative order of entries in each row of $A$. We denote this as

$$\text{rank}_{\text{rop}} A = \min\{\text{rank } B \mid B \in \mathbb{R}^{m \times n}, \text{ such that for all } i, j, k, \text{ if } A_{ij} > A_{ik} \text{ then } B_{ij} > B_{ik}\}.$$

In other words, if $A$ is a binary ground-truth relevance matrix, $\text{rank}_{\text{rop}} A$ is the minimum dimension necessary for any vector embedding model to return relevant documents before irrelevant ones for all queries. Alternatively, we might require that the scores of relevant documents can be cleanly separated from those of irrelevant ones by a threshold.

**Definition 2.** Given a binary matrix $A \in \{0, 1\}^{m \times n}$:

- The **row-wise thresholdable rank of** $A$ ($\text{rank}_{\text{rt}} A$) is the minimum rank of a matrix $B$ for which there exist row-specific thresholds $\{\tau_i\}_{i=1}^m$ such that for all $i, j$, $B_{ij} > \tau_i$ if $A_{ij} = 1$ and $B_{ij} < \tau_i$ if $A_{ij} = 0$.

- The **globally thresholdable rank of** $A$ ($\text{rank}_{\text{gt}} A$) is the minimum rank of a matrix $B$ for which there exists a single threshold $\tau$ such that for all $i, j$, $B_{ij} > \tau$ if $A_{ij} = 1$ and $B_{ij} < \tau$ if $A_{ij} = 0$.

**Remark 1.** This two-sided separation condition may be seen as slightly stronger than requiring $B_{ij} > \tau_i$ if and only if $A_{ij} = 1$, however since there are only finitely many elements of $B_{ij}$ we could always perturb the latter threshold by a sufficient number such that the two-sided condition holds.[11]

## D.2 THEORETICAL BOUNDS

For binary matrices, row-wise ordering/thresholding are equivalent notions of representation capacity.

**Proposition 1.** *For a binary matrix $A \in \{0, 1\}^{m \times n}$, we have that $\text{rank}_{\text{rop}} A = \text{rank}_{\text{rt}} A$.*

*Proof.* ($\leq$) Suppose $B$ and $\tau$ satisfy the row-wise thresholdable rank condition. Since $A$ is a binary matrix $A_{ij} > A_{ik}$ implies $A_{ij} = 1$ and $A_{ik} = 0$, thus $B_{ij} > \tau_i > B_{ik}$, and hence $B$ also satisfies the row-wise order-preserving condition.

($\geq$) Let $B$ satisfy the row-wise order-preserving condition, so $A_{ij} > A_{ik}$ implies $B_{ij} > B_{ik}$. For each row $i$, let $U_i = \{B_{ij} \mid A_{ij} = 1\}$ and $L_i = \{B_{ij} \mid A_{ij} = 0\}$. The row-wise order-preserving condition implies that every element of $U_i$ is greater than every element of $L_i$. We can therefore always find a threshold $\tau_i$ separating them (*e.g.* $\tau_i = (\max L_i + \min U_i)/2$ if both are non-empty, trivial otherwise). Thus $B$ is also row-wise thresholdable to $A$. $\square$

The notions we have described so far are closely related to the sign rank of a matrix, which we use in the rest of the paper to establish our main bounds.

**Definition 3** (Sign Rank). The sign rank of a matrix $M \in \{-1, 1\}^{m \times n}$ is the smallest integer $d$ such that there exists a rank $d$ matrix $B \in \mathbb{R}^{m \times n}$ whose entries have the same sign as those of $M$, i.e.

$$\text{rank}_{\pm} M = \min\{\text{rank } B \mid B \in \mathbb{R}^{m \times n} \text{ such that for all } i, j \text{ we have } \text{sign } B_{ij} = M_{ij}\}.$$

In what follows, we use $\mathbf{1}_n$ to denote the $n$-dimensional vector of ones, and $\mathbf{1}_{m \times n}$ to denote an $m \times n$ matrix of ones.

**Proposition 2.** *Let $A \in \{0, 1\}^{m \times n}$ be a binary matrix. Then $2A - \mathbf{1}_{m \times n} \in \{-1, 1\}^{m \times n}$ and*

$$\text{rank}_{\pm}(2A - \mathbf{1}_{m \times n}) - 1 \leq \text{rank}_{\text{rop}} A = \text{rank}_{\text{rt}} A \leq \text{rank}_{\text{gt}} A \leq \text{rank}_{\pm}(2A - \mathbf{1}_{m \times n})$$

---

[11]Without loss of generality, we may assume the thresholds in the above definitions are not equal to any elements of $B$ since we could increase the threshold of $\tau$ by a sufficiently small $\epsilon$ to preserve the inequality.

*Proof.* N.b. the equality was already shown in Proposition 1. We prove each inequality separately.

**1. $\text{rank}_{\mathbf{rt}} A \leq \text{rank}_{\mathbf{gt}} A$:** True by definition, since any matrix satisfying the globally thresholdable condition trivially satisfies a row-wise thresholdable condition with the same threshold for each row.

**2. $\text{rank}_{\mathbf{gt}} A \leq \text{rank}_{\pm}(2A - \mathbf{1}_{m \times n})$:** Let $B$ be any matrix whose entries have the same sign as $2A - \mathbf{1}_{m \times n}$, then

$$B_{ij} > 0 \iff 2A_{ij} - 1 > 0 \iff A_{ij} = 1.$$

Thus $B$ satisfies the globally thresholdable condition with a threshold of $0$.

**3. $\text{rank}_{\pm}(2A - \mathbf{1}_{m \times n}) - 1 \leq \text{rank}_{\mathbf{rt}} A$:** Suppose $B$ satisfies the row-wise thresholdable condition with minimal rank, so $\text{rank}_{\mathbf{rt}} A = \text{rank} B$ and there exists $\tau \in \mathbb{R}^m$ such that $B_{ij} > \tau_i$ if $A_{ij} = 1$ and $B_{ij} < \tau_i$ if $A_{ij} = 0$. Then the entries of $B - \tau \mathbf{1}_n^T$ have the same sign as $2A - \mathbf{1}_{m \times n}$, since $(B - \tau \mathbf{1}_n^T)_{ij} = B_{ij} - \tau_i$ and

$$B_{ij} - \tau_i > 0 \iff A_{ij} = 1 \iff 2A_{ij} - 1 > 0, \text{ and} \tag{3}$$

$$B_{ij} - \tau_i < 0 \iff A_{ij} = 0 \iff 2A_{ij} - 1 < 0. \tag{4}$$

Thus $\text{rank}_{\pm}(2A - \mathbf{1}_{m \times n}) \leq \text{rank}(B - \tau \mathbf{1}_n^T) \leq \text{rank}(B) + \text{rank}(\tau \mathbf{1}_n^T) = \text{rank}_{\mathbf{rt}} A + 1$.

Combining these gives the desired chain of inequalities. $\square$

### D.3 CONSEQUENCES

In the context of a vector embedding model, this provides a lower and upper bound on the dimension of vectors required to exactly capture a given set of retrieval objectives, in the sense of row-wise ordering, row-wise thresholding, or global thresholding. In particular, given some binary relevance matrix $A \in \{0, 1\}^{m \times n}$, we need at least $\text{rank}_{\pm}(2A - \mathbf{1}_{m \times n}) - 1$ dimensions to capture the relationships in $A$ exactly, and can always accomplish this in at most $\text{rank}_{\pm}(2A - \mathbf{1}_{m \times n})$ dimensions.

The cyclotomic polynomial construction presented in Alon et al. (1985) implies that any qrel matrix has sign-rank at most $2k$, where $k$ is the largest number of documents for a particular query. The construction results in unnormalized vectors, however this can be easily adapted to normalized vectors by using one additional dimension. In agreement with Theorem 1, this construction requires infinite precision in general, and is thus not feasible in practice.

### D.4 CORRELATION WITH MTEB

BEIR (used in MTEB v1) (Thakur et al., 2021; Muennighoff et al., 2022) has frequently been cited as something that embedding models have overfit to (Weller et al., 2025c; Thakur et al., 2025). We compare performance on LIMIT to BEIR in Figure 7. We see that performance is generally not correlated and that smaller models (like Arctic Embed) do worse on both, likely due to embedding dimension and pre-trained model knowledge.

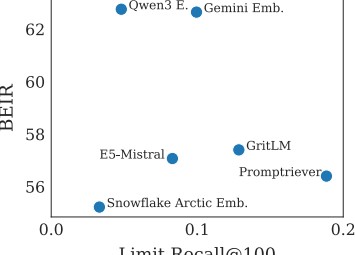

Figure 7: No obvious correlation between BEIR vs LIMIT.

## E LLM USAGE

LLMs were not used for any paper writing, only for coding help and title brainstorming.

## F METRICS MEASURING QREL GRAPH DENSITY

We show two metrics that treat the qrel matrix as a graph and show that LIMIT has unique properties compared to standard IR datasets (Table 2). We call these metrics Graph Density and Average Query Strength and describe them below.

**Graph Density** We use the qrel matrix to construct the graph, where nodes are documents and an edge exists between two documents if they are both relevant to at least one common query.

For a given graph $G = (V, E)$ with $V$ being the set of nodes and $E$ being the set of edges, the graph density is defined as the ratio of the number of edges in the graph to the maximum possible number of edges. For an undirected graph, the maximum possible number of edges is $\frac{|V|(|V|-1)}{2}$. Thus, the density $\rho$ is calculated as:

$$\rho = \frac{|E|}{\frac{|V|(|V|-1)}{2}} = \frac{2|E|}{|V|(|V|-1)}$$

This metric indicates how connected the graph is; a density of 1 signifies a complete graph (all possible edges exist), while a density close to 0 indicates a sparse graph. For a qrel dataset, the

**Average Query Strength**   In a query-query graph where nodes are queries and edges represent similarity between queries (e.g., Jaccard similarity of their relevant documents), the *strength* of a query node $i$, denoted $s_i$, is defined as the sum of the weights of all edges incident to it. If $w_{ij}$ is the weight of the edge between query $i$ and query $j$, and $N(i)$ is the set of neighbors of query $i$, then the strength is:

$$s_i = \sum_{j \in N(i)} w_{ij}$$

The Average Query Strength $\bar{s}$ is the mean of these strengths across all query nodes in the graph:

$$\bar{s} = \frac{1}{|V_Q|} \sum_{i \in V_Q} s_i$$

where $V_Q$ is the set of all query nodes in the graph. This metric provides an overall measure of how strongly connected queries are to each other on average within the dataset, based on their shared relevant documents.

**Comparisons to other datasets**   We compare with standard IR Datasets such as NQ (Kwiatkowski et al., 2019), HotpotQA (Yang et al., 2018), and SciFact (Wadden et al., 2020). We also show an instruction-following dataset, FollowIR Core17 (Weller et al., 2024a). For all datasets, we use the test set only. The results in Table 2 show that LIMIT has significantly higher values for both of these metrics (i.e. 28 for query similarity compared to 0.6 or lower for the others).

Table 2: Metrics measuring the density of the qrel matrix. We see that LIMIT is significantly higher than other datasets, but that the closest are instruction-following datasets such as Core17 from FollowIR. Our empirical ablations suggest (although cannot definitively prove) that datasets with higher values here will be harder for retrieval models to represent.

| Dataset Name | Graph Density | Average Query Strength |
|---|---|---|
| NQ | 0 | 0 |
| HotPotQA | 0.000037 | 0.1104 |
| SciFact | 0.001449 | 0.4222 |
| FollowIR Core17 | 0.025641 | 0.5912 |
| LIMIT | 0.085481 | 28.4653 |

# G   TABLE FORMS OF FIGURES

In this section we show the table form of various figures. For Figure 3 it is Table 5, Figure 4 in Table 4, Figure 2 in Table 6, and Figure 5 in Table 3.

| Split | Dim | Recall@2 | Recall@10 | Recall@100 |
|-------|------|----------|-----------|------------|
| Test | 32 | 85.5 | 98.4 | 100.0 |
| Test | 64 | 90.4 | 98.7 | 100.0 |
| Test | 128 | 93.1 | 99.5 | 99.9 |
| Test | 256 | 94.2 | 99.7 | 100.0 |
| Test | 384 | 95.6 | 99.6 | 100.0 |
| Test | 512 | 94.0 | 99.5 | 99.9 |
| Test | 768 | 96.1 | 99.8 | 100.0 |
| Test | 1024 | 96.5 | 99.8 | 100.0 |
| Train | 32 | 0.0 | 0.0 | 0.0 |
| Train | 64 | 0.1 | 0.3 | 2.2 |
| Train | 128 | 0.2 | 0.7 | 3.1 |
| Train | 256 | 0.0 | 0.0 | 0.4 |
| Train | 384 | 1.1 | 2.7 | 8.3 |
| Train | 512 | 0.7 | 2.3 | 9.8 |
| Train | 768 | 0.7 | 2.4 | 9.9 |
| Train | 1024 | 1.0 | 2.8 | 11.2 |

Table 3: Fine-tuning results in table form. See Figure 5 for the comparable plot.

| Model | Dim | Recall@2 | Recall@10 | Recall@20 |
|---|---|---|---|---|
| BM25 | default | 97.8 | 100.0 | 100.0 |
| E5-Mistral 7B | 32 | 7.9 | 32.6 | 56.2 |
| E5-Mistral 7B | 64 | 10.2 | 37.0 | 60.3 |
| E5-Mistral 7B | 128 | 14.5 | 41.9 | 65.9 |
| E5-Mistral 7B | 256 | 15.3 | 45.9 | 69.7 |
| E5-Mistral 7B | 512 | 22.2 | 54.7 | 74.8 |
| E5-Mistral 7B | 768 | 21.6 | 57.5 | 79.2 |
| E5-Mistral 7B | 1024 | 24.5 | 60.5 | 80.0 |
| E5-Mistral 7B | 2048 | 28.9 | 66.3 | 83.2 |
| E5-Mistral 7B | 3072 | 29.9 | 67.8 | 85.3 |
| E5-Mistral 7B | 4096 | 29.5 | 68.1 | 85.2 |
| GTE-ModernColBERT | default | 83.5 | 97.6 | 99.1 |
| GritLM 7B | 32 | 7.8 | 33.5 | 56.3 |
| GritLM 7B | 64 | 9.4 | 35.9 | 59.6 |
| GritLM 7B | 128 | 14.2 | 42.7 | 64.9 |
| GritLM 7B | 256 | 17.3 | 46.2 | 68.3 |
| GritLM 7B | 512 | 21.8 | 55.6 | 76.7 |
| GritLM 7B | 768 | 23.8 | 58.1 | 80.1 |
| GritLM 7B | 1024 | 26.2 | 61.4 | 80.1 |
| GritLM 7B | 2048 | 33.0 | 69.1 | 86.2 |
| GritLM 7B | 3072 | 36.3 | 72.9 | 89.9 |
| GritLM 7B | 4096 | 38.4 | 75.4 | 90.5 |
| Promptriever Llama3 8B | 32 | 6.1 | 31.4 | 56.0 |
| Promptriever Llama3 8B | 64 | 8.9 | 35.8 | 62.3 |
| Promptriever Llama3 8B | 128 | 13.7 | 44.5 | 67.6 |
| Promptriever Llama3 8B | 256 | 18.5 | 52.1 | 74.1 |
| Promptriever Llama3 8B | 512 | 27.0 | 61.8 | 81.7 |
| Promptriever Llama3 8B | 768 | 35.5 | 69.0 | 84.7 |
| Promptriever Llama3 8B | 1024 | 38.0 | 73.5 | 89.1 |
| Promptriever Llama3 8B | 2048 | 46.2 | 83.6 | 94.2 |
| Promptriever Llama3 8B | 3072 | 49.2 | 87.3 | 96.6 |
| Promptriever Llama3 8B | 4096 | 54.3 | 90.0 | 97.7 |
| Qwen3 Embed | 32 | 8.3 | 30.6 | 53.9 |
| Qwen3 Embed | 64 | 9.4 | 35.5 | 57.6 |
| Qwen3 Embed | 128 | 11.6 | 38.3 | 60.8 |
| Qwen3 Embed | 256 | 14.3 | 41.6 | 63.8 |
| Qwen3 Embed | 512 | 16.1 | 43.7 | 66.0 |
| Qwen3 Embed | 768 | 17.2 | 45.3 | 69.3 |
| Qwen3 Embed | 1024 | 17.8 | 48.7 | 70.3 |
| Qwen3 Embed | 2048 | 19.5 | 51.5 | 72.4 |
| Qwen3 Embed | 3072 | 19.3 | 52.8 | 73.3 |
| Qwen3 Embed | 4096 | 19.0 | 52.3 | 73.8 |
| Gemini Embed | 2 | 4.2 | 23.0 | 45.5 |
| Gemini Embed | 4 | 4.2 | 21.9 | 46.0 |
| Gemini Embed | 8 | 4.9 | 23.2 | 47.0 |
| Gemini Embed | 16 | 5.2 | 24.7 | 47.5 |
| Gemini Embed | 32 | 6.3 | 25.2 | 50.6 |
| Gemini Embed | 64 | 6.9 | 30.6 | 55.0 |
| Gemini Embed | 128 | 7.7 | 37.0 | 62.9 |
| Gemini Embed | 256 | 14.6 | 46.9 | 69.7 |
| Gemini Embed | 512 | 23.3 | 58.4 | 77.9 |
| Gemini Embed | 768 | 28.8 | 67.5 | 84.5 |
| Gemini Embed | 1024 | 31.8 | 69.9 | 86.1 |
| Gemini Embed | 2048 | 31.9 | 70.3 | 87.1 |
| Gemini Embed | 3072 | 33.7 | 72.4 | 87.9 |
| Snowflake Arctic L | 32 | 8.3 | 30.3 | 53.8 |
| Snowflake Arctic L | 64 | 9.0 | 35.4 | 58.5 |
| Snowflake Arctic L | 128 | 12.7 | 41.3 | 65.1 |
| Snowflake Arctic L | 256 | 16.0 | 48.2 | 72.6 |
| Snowflake Arctic L | 512 | 16.7 | 51.3 | 74.1 |
| Snowflake Arctic L | 768 | 17.9 | 53.5 | 74.6 |
| Snowflake Arctic L | 1024 | 19.4 | 54.9 | 76.0 |
| Snowflake Arctic L | 2048 | 19.4 | 54.9 | 76.0 |
| Snowflake Arctic L | 3072 | 19.4 | 54.9 | 76.0 |
| Snowflake Arctic L | 4096 | 19.4 | 54.9 | 76.0 |

Table 4: Results for the LIMIT small version. See comparable Figure 4.

| Model | Dim | Recall@2 | Recall@10 | Recall@100 |
|---|---|---|---|---|
| E5-Mistral 7B | 32 | 0.0 | 0.0 | 0.5 |
| E5-Mistral 7B | 64 | 0.0 | 0.1 | 0.4 |
| E5-Mistral 7B | 128 | 0.1 | 0.3 | 1.0 |
| E5-Mistral 7B | 256 | 0.4 | 0.9 | 1.9 |
| E5-Mistral 7B | 512 | 0.7 | 1.3 | 3.8 |
| E5-Mistral 7B | 768 | 0.9 | 1.7 | 4.3 |
| E5-Mistral 7B | 1024 | 0.9 | 1.8 | 5.9 |
| E5-Mistral 7B | 2048 | 1.0 | 1.9 | 6.8 |
| E5-Mistral 7B | 3072 | 1.3 | 2.0 | 7.7 |
| E5-Mistral 7B | 4096 | 1.3 | 2.2 | 8.3 |
| Snowflake Arctic L | 32 | 0.0 | 0.1 | 0.6 |
| Snowflake Arctic L | 64 | 0.2 | 0.4 | 1.7 |
| Snowflake Arctic L | 128 | 0.1 | 0.3 | 1.8 |
| Snowflake Arctic L | 256 | 0.2 | 0.8 | 2.5 |
| Snowflake Arctic L | 512 | 0.3 | 1.0 | 2.5 |
| Snowflake Arctic L | 768 | 0.4 | 1.1 | 3.1 |
| Snowflake Arctic L | 1024 | 0.4 | 0.8 | 3.3 |
| Snowflake Arctic L | 2048 | 0.4 | 0.8 | 3.3 |
| Snowflake Arctic L | 3072 | 0.4 | 0.8 | 3.3 |
| Snowflake Arctic L | 4096 | 0.4 | 0.8 | 3.3 |
| GritLM 7B | 32 | 0.0 | 0.0 | 0.8 |
| GritLM 7B | 64 | 0.0 | 0.1 | 0.3 |
| GritLM 7B | 128 | 0.1 | 0.3 | 1.3 |
| GritLM 7B | 256 | 0.1 | 0.4 | 2.8 |
| GritLM 7B | 512 | 0.6 | 1.8 | 6.5 |
| GritLM 7B | 768 | 1.5 | 3.1 | 8.7 |
| GritLM 7B | 1024 | 1.8 | 3.5 | 10.6 |
| GritLM 7B | 2048 | 2.3 | 4.3 | 11.8 |
| GritLM 7B | 3072 | 2.0 | 4.3 | 12.9 |
| GritLM 7B | 4096 | 2.4 | 4.1 | 12.9 |
| Promptriever Llama3 8B | 32 | 0.0 | 0.0 | 0.1 |
| Promptriever Llama3 8B | 64 | 0.0 | 0.0 | 0.3 |
| Promptriever Llama3 8B | 128 | 0.0 | 0.1 | 0.6 |
| Promptriever Llama3 8B | 256 | 0.2 | 0.4 | 1.8 |
| Promptriever Llama3 8B | 512 | 0.6 | 1.4 | 5.4 |
| Promptriever Llama3 8B | 768 | 1.3 | 3.1 | 8.7 |
| Promptriever Llama3 8B | 1024 | 2.1 | 4.4 | 12.8 |
| Promptriever Llama3 8B | 2048 | 3.2 | 6.5 | 18.1 |
| Promptriever Llama3 8B | 3072 | 2.9 | 6.3 | 17.8 |
| Promptriever Llama3 8B | 4096 | 3.0 | 6.8 | 18.9 |
| Qwen3 Embed | 32 | 0.0 | 0.1 | 1.1 |
| Qwen3 Embed | 64 | 0.0 | 0.2 | 1.0 |
| Qwen3 Embed | 128 | 0.3 | 0.4 | 1.8 |
| Qwen3 Embed | 256 | 0.4 | 0.8 | 3.2 |
| Qwen3 Embed | 512 | 0.6 | 1.3 | 3.3 |
| Qwen3 Embed | 768 | 0.7 | 1.5 | 3.8 |
| Qwen3 Embed | 1024 | 0.7 | 1.6 | 4.6 |
| Qwen3 Embed | 2048 | 0.9 | 1.7 | 4.7 |
| Qwen3 Embed | 3072 | 0.8 | 1.6 | 4.8 |
| Qwen3 Embed | 4096 | 0.8 | 1.8 | 4.8 |
| Gemini Embed | 2 | 0.0 | 0.0 | 0.1 |
| Gemini Embed | 4 | 0.0 | 0.0 | 0.0 |
| Gemini Embed | 8 | 0.0 | 0.0 | 0.0 |
| Gemini Embed | 16 | 0.0 | 0.0 | 0.0 |
| Gemini Embed | 32 | 0.0 | 0.0 | 0.0 |
| Gemini Embed | 64 | 0.0 | 0.0 | 0.3 |
| Gemini Embed | 128 | 0.0 | 0.1 | 0.3 |
| Gemini Embed | 256 | 0.0 | 0.1 | 1.2 |
| Gemini Embed | 512 | 0.2 | 1.1 | 3.6 |
| Gemini Embed | 768 | 0.9 | 2.5 | 7.6 |
| Gemini Embed | 1024 | 1.3 | 2.7 | 8.1 |
| Gemini Embed | 2048 | 1.5 | 3.1 | 8.5 |
| Gemini Embed | 3072 | 1.6 | 3.5 | 10.0 |
| GTE-ModernColBERT | default | 23.1 | 34.6 | 54.8 |
| BM25 | default | 85.7 | 90.4 | 93.6 |

Table 5: Results on LIMIT. See comparable Figure 3.

| $d$ | Critical-$n$ |
|---|---|
| 4 | 10 |
| 5 | 14 |
| 6 | 19 |
| 7 | 24 |
| 8 | 28 |
| 9 | 32 |
| 10 | 36 |
| 11 | 42 |
| 12 | 47 |
| 13 | 54 |
| 14 | 62 |
| 15 | 70 |
| 16 | 79 |
| 17 | 89 |
| 18 | 99 |
| 19 | 109 |
| 20 | 120 |
| 21 | 132 |
| 22 | 144 |
| 23 | 157 |
| 24 | 170 |
| 25 | 184 |
| 26 | 198 |
| 27 | 213 |
| 28 | 229 |
| 29 | 245 |
| 30 | 261 |
| 31 | 278 |
| 32 | 296 |
| 33 | 314 |
| 34 | 333 |
| 35 | 352 |
| 36 | 372 |
| 37 | 392 |
| 38 | 413 |
| 39 | 434 |
| 40 | 460 |
| 41 | 484 |
| 42 | 505 |
| 43 | 545 |
| 44 | 605 |
| 45 | 626 |

Table 6: Critical Values of n for different d values in the Free Embedding optimization experiments. See Figure 2 for the corresponding figure.

| Model | BEIR | LIMIT R@100 |
|---|---|---|
| Snowflake Arctic | 55.22 | 3.3 |
| Promptriever | 56.40 | 18.9 |
| E5-Mistral | 57.07 | 8.3 |
| GritLM | 57.40 | 12.9 |
| Gemini Embed | 62.65 | 10.0 |
| Qwen3 Embed | 62.76 | 4.8 |

Table 7: BEIR vs LIMIT results. See Figure 7 for the comparable plot.