# OpenReview forum: "On the Theoretical Limitations of Embedding-Based Retrieval"
_ICLR.cc/2026/Conference — ICLR 2026 Poster_

### Official Review · Reviewer_8eB8 · 2025-10-25

**Soundness:** 2
**Presentation:** 2
**Contribution:** 2
**Rating:** 4
**Confidence:** 4

**Summary:**

With the ubiquitous use of embedding models, the authors look at theoretical limits for use of vector embeddings which cannot be overcome with increase of training data. They setup their mathematical propositions and hypothesize from the same about limitations of dense embedding models for retrieval tasks. They create a new dataset and use that to experiment. Their results indicate that the scores can be extremely poor for their settings even after training.

**Strengths:**

It is important in today's time with the ubiquitous use of embedding models, to carefully understand their limitations. The authors take both a theoretical and experimental approach to this very important problem.

1. The mathematical  formulation of the problem is strong.
2. The authors create a new dataset LIMIT  and use that for their work.
3. The authors establish that for tasks defined in LIMIT dense embedding models perform extremely poorly.

**Weaknesses:**

The experimental results are interesting. However, one must ask a pertitent question - are the authors being too pessimistic in their overall claims by creating a dataset and choosing a task which are incompatible with each other? Fundamentally, they have a dataset which is not semantic based and is a word matched based and they have an approach via dense vector embedding search which is unsuitable for this. This mismatch of choice clearly shows by the significant superiority of colbert and the near perfect match of BM25. This indicates experimenting with a wrong approach for a problem - like the cliched "if the only tool you have is a hammer .." .

The fact that BM25 is hard to outperform on keyword heavy tasks is well researched in the literature [1-4] are examples from a quick search. I am sure a more focussed search will bring up more results.

So the question that one wonders after reading the paper - is are the claims in the abstract "Our work shows the limits of
embedding models under the existing single vector paradigm and calls for future research to develop new techniques that can resolve this fundamental limitation" are substantiated only by misplaced mapping of experiment and dataset. Their claim on LIMIT being a realistic dataset is valid but it is NOT a dataset well suited for the approach chosen. If this paper purpose was to establish to the practioner / reseacher that do not use embedding models blindly for any task it will be a very well experimented result despite the citations below. However the authors claims are far more stronger which therefore need a stronger and better experimental paradigm.

Their extremely poor numbers of Recall are also an indication of a wrong technique to a problem and is potentially misleading to a reader who may not be as dilligent - one cannot deny the success of embedding models (the authors do not) but their results leave one wondering what went wrong.

[1] BEIR: A Heterogeneous Benchmark for Zero-shot Evaluation of Information Retrieval Models — Thakur et al., 2021.

[2] Match Your Words! / A Thorough Examination on Lexical Matching in Neural IR — Formal (T. Formal), 2021/2022.

[3] Salient Phrase Aware Dense Retrieval — “Can a Dense Retriever Imitate a Sparse One?” — Chen et al., 2021/2022 (SPAR).

[4] A Thorough Examination on Zero-shot Dense Retrieval](https://aclanthology.org/2023.findings-emnlp.1057/) (Ren et al., Findings 2023)

Minor comment:-

I note here details on writing and clarity.

1. the d in figure 2 is obtained by the hp-tuning. d is also used in definition of rank_{rt}. Line 166 defines d to be the dimension. I understand they are related but the d in figure 2 is effectively the din line 166.  maybe a different variable would have been easier.
2. line 356 refers to appendix C but this does not exist in the pdf.
3. "1000 queries to maintain statistical significance while still being fast to evaluate" - statistical significance is a bit loosely used.
4. "For example, in the full setting models struggle to reach even 20% recall@100 and in the 46 document version models cannot solve the task even with recall@20" - I cannot find two sets of results.
5. Line 457 mentions a Figure 5 which is not present in the document.
6. I struggled deciphering the colours in figure 3 and which model is colbert and which is BM25.

**Questions:**

1. I understand experimentally it is not possible to extend the experiments in Figure 2 beyond the numbers given, but are there any theoretical or empirical reasons to expect the equation can be extended to hundreds of times from about 50 to 4096?
2. Will LIMIT be/is publicly available ?
3. please address the concerns in the weaknesses section.

---

> ### Author Response · Authors · 2025-11-20
>
> Thank you for recognizing that our “mathematical formulation of the problem is strong”!
>
> > are the authors being too pessimistic in their overall claims by creating a dataset and choosing a task which are incompatible with each other?
>
> **There may be a misunderstanding: we agree with you!** In fact, we designed this dataset to be difficult for dense vectors but not so for lexical search due to the high dimensionality requirement (lexical models use more dimensions generally)! After all, one might wonder why neural models fail to adapt to these tasks as well as their BM25 counterparts despite the task being very easy.. This allows us to relate these practical findings to their theoretical underpinnings.
>
> Our instantiation of LIMIT is the simplest way to illustrate our theoretical results and was designed with lexical overlap to be easy to understand. However, one could replace the lexical-heavy version we used with a different abstraction, as long as the underlying many-to-many mapping occurs in the qrels.
>
> Upon your suggestion we created a version of LIMIT-small that replaces all items in the corpus with their synonyms, reducing the amount of lexical overlap. We ask Gemini 2.5 Pro to come up with synonyms that don’t match any other existing synonyms or original items, using either scientific names, similar meanings, or if necessary hypernyms. This creates a mapping like “glasses” -> “spectacles”, etc. We note that it doesn't remove all overlap due to items like “Scuba Diving” -> “Underwater Diving”.  However, when we repeat the experiment on LIMIT-small (synonyms) and compare to LIMIT-small we see similar results for recall@2 scores:
>
> | Model                      | LIMIT-Small | LIMIT-Small Synonyms |
> | -------------------------- | ----------- | -------------------- |
> | **GTE-ModernColBERT**      | 83.5        | 25.6                 |
> | **BM25**                   | 97.8        | 10.6                 |
> | **Snowflake Arctic L**     | 19.4        | 8.5                  |
> | **E5-Mistral 7B**          | 29.5        | 15.1                 |
> | **GritLM 7B**              | 38.4        | 14.3                 |
> | **Promptriever Llama3 8B** | 54.3        | 12.8                 |
> | **Qwen3 Embed**            | 19.0        | 11.6                 |
>
> We see that all models have a harder time, but notably BM25 drops to be worse than nearly all other embedding models. BM25 performance is non-zero due to some overlap in synonyms and would go down if we continued to refine this dataset to remove lexical overlap. **Thus, as we mention in Section 5.3, sparse models are not a full solution to this problem and new approaches are needed**.
>
> We hope the inclusion of this experiment in the paper will help resolve your concern: we do not mean to suggest that BM25 is the solution, as it has many drawbacks itself.
>
> > I understand experimentally it is not possible to extend the experiments in Figure 2 beyond the numbers given, but are there any theoretical or empirical reasons to expect the equation can be extended to hundreds of times from about 50 to 4096?
>
> The free embeddings are purely empirical as you mention. However, the link between theory and practice aligns with the results seen in the free embeddings, as extra confirmation.
>
> > Will LIMIT be/is publicly available ?
>
> Yes, it has already been open-sourced. If you wish to examine it during the review period, we link an anonymous version here: https://anonymous.4open.science/r/limit-C689/README.md.
>
> > I note here details on writing and clarity.
>
> Thank you for these notes, we will update the paper. The 46 document version of LIMIT  was placed in the appendix (Figure 5) due to space. We will bring it up with the extra page allotted.

---

> ### Comment · Reviewer_8eB8 · 2025-11-20
>
> Thank you for the response and the additional experiments. I would like to ask a few follow-up questions just to be sure I understand it fully.
>
> 1. Your manuscript does not have a table but if I look at Figure Figure 3a, the Recall@2 for models other than Colbert and BM25 looks like to be  less than 10%. Figure 5 is not on the manuscript to compare - I assume the numbers you have quoted in your fresh experiments pertain to the missing Figure 5.
> 2.  This however raises a question - while it is understandable, indeed expected, that BM25 takes a massive hit,  how do we understand why the dense embeddings also take a 50% hit from already poor scores? Should they not have maintained similar performance given the semantic closeness? The only explanation I can think of is that the extremely small and synthetic dataset does not map to any of its training corpus and hence it is so poor. Would like to understand your thoughts on the same, especially inline with the next question.
> 3.  I do not think you will dispute the fact that embedding models have been tremendously successful in both academic and industry solutions for retrieval etc. The unnnaturally low scores on your data raises a concern that this does not make anyone the wiser on what are the actual limitations - the dataset being not matched to the task measurements on it are non-interpretable on the limitations of embedding based retrieval. Your additional experiments on synonyms establishes the fact that there is no semantic understanding being utilized for this task - which is basically a fancy but inefficient way to do a grep (I just checked it out - thanks for sharing the data). What can the community take away from this work - what are reasonable limitations observed by experimentation since these numbers are unsuitable to be interpreted for limitations for either research or practice.
> ```
> % grep "Slide Rules" corpus.jsonl
> {"_id": "Geneva Durben", "title": "", "text": "Geneva Durben likes Quokkas, River Otters, Tapirs, Asymmetry, Snow Leopards, Hydrangeas, Crocheting, Symphonies, Bassoons, Pangolins, Limes, Mosaic, Creativity, Licorice, Spinach, Chairs, Horror Literature, Scuba Diving, Glaciers, Cartography, Alternative Rock, Oceanic Trenches, Barley, Cheerios, Sapphires, Parakeets, The Aztec Empire, Traveling, Candle Making, Shasta Daisies, Patterns, Disco Music, White Pepper, Slide Rules, Halibut, Giant Isopods, Praying Mantises, Fables, Alligators, Caracals, Joshua Trees, Pansies, Soy Sauce, Cards Against Humanity and Elm Trees."}
> {"_id": "Armand Schweda", "title": "", "text": "Armand Schweda likes Eyeglasses, Mechanics, Mandolins, Barbershop quartets, Soap Making, Gecko Lizards, Herbal Tea, Volleyball, Western Movies, Optimism, Armadillos, Rubik's Cubes, Quiches, Vegan Cheese Pizza, Quilts, Saiga Antelopes, Cloves, Tango Dancing, Thin Mints, Dragon Fruits, Patience, Cheesecake, Wombats, Sausages, Cool Breezes, Satisfaction, Editors, Ukuleles, Sea Lions, Disc Golf, Community, Cassette Players, Smoked Paprika, Slide Rules, Minimalism, Architecture, Televisions, Soccer, Baking Powder, Starfish, Monarch Butterflies, Futurism, Amethyst, Goat Cheese and Walruses."}
>
> % grep "Goat Cheese" corpus.jsonl
> {"_id": "Armand Schweda", "title": "", "text": "Armand Schweda likes Eyeglasses, Mechanics, Mandolins, Barbershop quartets, Soap Making, Gecko Lizards, Herbal Tea, Volleyball, Western Movies, Optimism, Armadillos, Rubik's Cubes, Quiches, Vegan Cheese Pizza, Quilts, Saiga Antelopes, Cloves, Tango Dancing, Thin Mints, Dragon Fruits, Patience, Cheesecake, Wombats, Sausages, Cool Breezes, Satisfaction, Editors, Ukuleles, Sea Lions, Disc Golf, Community, Cassette Players, Smoked Paprika, Slide Rules, Minimalism, Architecture, Televisions, Soccer, Baking Powder, Starfish, Monarch Butterflies, Futurism, Amethyst, Goat Cheese and Walruses."}
> {"_id": "Darwyn Raio", "title": "", "text": "Darwyn Raio likes Waffles, Patios, Pangolins, Juniper Trees, Sweaters, Mural Painting, Ancient Egypt, Volcanoes, Hummus, Plateaus, Skip-Bo, the San Diego Padres, Peacocks, Euphoria, Umbrellas, Black Forest Cakes, Saxophones, Beachcombing, The Roaring Twenties, Grapefruits, The Information Age, Singing, Lollipops, Forgiveness, Mountain Biking, Hazelnut Flavor, Shower Curtains, Chives, Paleontologists, Economics, Anacondas, Origami, Bison, Bleeding Hearts, Conga Drums, Black Garden Ants, Frilled Sharks, Goat Cheese, the Pittsburgh Pirates, Twinkies, Caramel Candies, Acacia Trees, Photorealism, Beignets and Cobras."}
> ```
> 4. Your mathematical formulation is strong but is not amenable to any reasonable limits for semantic retrieval problems, as far as I can understand. I mean one cannot in any problem say that given the dataset and the dimension of the embeddings this is the max expected limit of retrieval performance no matter how "simplistic" the queries are.
>
> Would be happy to be corrected if I am misunderstanding any aspect of the experiments.

---

> > ### Comment · Reviewer_8eB8 · 2025-11-20
> >
> > Am I correct in understanding that you are embedding the whole sentence "Armand Schweda likes Eyeglasses, Mechanics, Mandolins, Barbershop quartets, Soap Making, Gecko Lizards, Herbal Tea, Volleyball, Western Movies, Optimism, Armadillos, Rubik's Cubes, Quiches, Vegan Cheese Pizza, Quilts, Saiga Antelopes, Cloves, Tango Dancing, Thin Mints, Dragon Fruits, Patience, Cheesecake, Wombats, Sausages, Cool Breezes, Satisfaction, Editors, Ukuleles, Sea Lions, Disc Golf, Community, Cassette Players, Smoked Paprika, Slide Rules, Minimalism, Architecture, Televisions, Soccer, Baking Powder, Starfish, Monarch Butterflies, Futurism, Amethyst, Goat Cheese and Walruses." and trying to retrieve it via a question like "Who likes eyeglasses" - the effectiveness of retrieving based on individual keywords using dense retrieval is known to be low and is not unexpected.
> >
> > On the other hand if you had split the sentence into individual sentences like "Armand Schweda likes Eyeglasses," , "Armand Schweda likes Mechanics" etc I am pretty sure even with dense retrieval we will get quite good scores and fall of accuracy using synonyms would also be much less dramatic. Would it be possible to evaluate this on the LIMIT-small?

---

> > > ### Author Response · Authors · 2025-11-20
> > >
> > > > On the other hand if you had split the sentence into individual sentences like "Armand Schweda likes Eyeglasses,"
> > >
> > > Sadly we cannot, as the proof requires all combinations of document to be returnable given a single query. This is why we chose the simplest setting, where each document has attributes that can be used to return any two unique documents given a single query.
> > >
> > > In your proposed setting, performance would improve, but this property would not exist as every document would not be connected to every other document.

---

> > ### Author Response · Authors · 2025-11-20
> >
> > Thank you for your response! We realize there are two misunderstandings which we hope to clear up!
> >
> > ### 1. Missing figures
> >
> > Somehow the appendix didn't upload in the original submission and thus prevented you from seeing Figure 5. Our apologies, we are not sure how that happened. All the figures and table forms are now present in the recent update if you click on the pdf again.
> >
> > Figure 5 was the LIMIT-small referenced in the main text. It is not the one referenced in Figure 3. Figure 3 is the full dataset which is harder for these models given the larger corpus (50k). If you look at Figure 5, you will see that models can do well with higher recall numbers given that the corpus is significantly smaller.
> >
> > ### 2: Questions about low scores
> >
> > It seems the 2nd main misunderstanding is about the low absolute scores of the models. We agree with you on all accounts about it being known that BM25 is a better lexical search model and that dense models struggle with it compared to lexical models. However, we adopted this instantiation of LIMIT (and the bias that comes with it) as a way to express our main contributions in the most simple manner: which is that embeddings are limited by their dimensionality.
> >
> > Regardless of the absolute score of the dense models, the crucial insight is that performance increases as dimensionality increases. That is the main contribution of our work, is that embedding models are limited by this embedding dimension. This starts with the proof that shows that models can’t represent all combinations, and the free embeddings which show that even in the optimal optimization case they can’t do it, and finally the instantiation of the dataset provides empirical evidence of our theoretical results showing improved results with higher dimensionality.
> >
> > Do you agree with the above paragraph as our contribution? We will answer more specific questions below, but this this may be the core misunderstanding between us.

---

> ### Author Response · Authors · 2025-11-20
>
> More detailed response as promised:
>
> > how do we understand why the dense embeddings also take a 50% hit from already poor scores
>
> This is to be expected that they take a large performance hit. It is well known in previous work that paraphrasing queries makes it harder for dense embeddings also [1-3], see roughly 50% drops in [1] also for example. Intuitively, you can view this as it’s easier to do an exact match using the same word embedding rather than trying to match similar but different tokens.
>
> We apologize if this new experiment caused additional confusion: the main point of this new experiment was just to show that BM25 is not a panacea. However, paraphrasing will be more difficult than exact match for all models.
>
> > I do not think you will dispute the fact that embedding models have been tremendously successful
>
> We do not disagree and in fact state this ourselves almost word for word in line 90 in the intro. They have been very successful! We are trying to help make sure they stay successful but showing their limitations to this next generation of any relevance definition and any instruction queries.
>
> > Your additional experiments on synonyms establishes the fact that there is no semantic understanding being utilized for this task - which is basically a fancy but inefficient way to do a grep
>
> Indeed LIMIT (non-synonyms) can be solved with grep and we state this ourselves in footnote 9.
>
> However, it is hard to argue that dense models shouldn’t be able to handle exact lookups also (in fact, it should be easier than a synonym task as the same word can be embedded).  As for one timely example of why this matters in real life, see the recent blog post from Cognition AI on how vector embeddings perform poorly on code due to their lack of exact match recall: https://cognition.ai/blog/swe-grep
>
> We agree that performance is intrinsically linked to the format of the task and that using lexical exact match introduces bias. However, any instantiation we create will come with a bias. We chose the bias that is the most simple to illustrate our point and can easily be seen by checking the curves of dimensionality in our plots. The proof, free embeddings, and empirical results all indicate the same phenomenon.
>
> > Your mathematical formulation is strong but is not amenable to any reasonable limits for semantic retrieval problems, as far as I can understand. I mean one cannot in any problem say that given the dataset and the dimension of the embeddings this is the max expected limit of retrieval performance no matter how "simplistic" the queries are
>
> Perhaps there is a misunderstanding. We do not claim (or even try) to predict the performance scores. We simply aim to show that models are limited by this and cannot retrieval all combinations. That is our mathematical formulation. Please let us know if you think that is not true.
>
> #### References
> [1] On the Robustness of Generative Retrieval Models: An Out-of-Distribution Perspective by Liu et. al. 2023, see Table 4 “paraphrasing” column
> [2] Pipelines with Query Variation Generators by Penha et. al. 2023 (Table 5 shows drops from paraphrases)
> [3] Investigating the Robustness of Retrieval-Augmented Generation at the
> Query Leve by Perçin et. al. 2025  (see FIgure 2, also Table 4 for retriever specific numbers)

---

> > ### Comment · Reviewer_8eB8 · 2025-11-21
> >
> > Thank you for your detailed comments.
> >
> > 1. I understand the constraint of the experiment - thank you for clarifying.
> >
> > 2. I can see the images - thank you. I was confused even in the earlier response on why you kept on referring to images which don't exist.
> >
> > 3. I don't disagree with the fact that performance increases when dimensionality increases. You prove mathematically but this is sort of expected - I am not aware offhand of any proofs but I do not think anyone in the field would doubt that statement even if you had stated without proof.
> >
> > 4. ```Perhaps there is a misunderstanding. We do not claim (or even try) to predict the performance scores. We simply aim to show that models are limited by this and cannot retrieval all combinations. That is our mathematical formulation. Please let us know if you think that is not true.```
> >
> > I know you do not claim but without which I still am worried if this paper's scores confuses the reader. Given the very good performance of embedding models and suddenly they see they are really struggling. One should not think that the entire embedding based retrieval is all broken. I wonder whether the challenges of this dataset towards embedding based retrieval should be made more clearer to the reader and one should not read the absolute numbers to intrepet anything but rather compare for different embedding dimensions? If this is clear already in the manuscript, please highlight the sentence(s) if not would you be willing to highlight it.
> >
> > Thank you for engaging in a constructive way. If you can comment on the last question above , I would be happy to revise my scores.

---

> > > ### Author Response · Authors · 2025-11-22
> > >
> > > Thank you for your engagement and for your patience as we figured out the appendix issue!
> > >
> > > > You prove mathematically but this is sort of expected - I am not aware offhand of any proofs but I do not think anyone in the field would doubt that statement even if you had stated without proof.
> > >
> > > We appreciate that you think this is expected and thus intuitive! Sadly this is not universal, and even Reviewer i6hx in this review process thinks differently. In our experience people are split: some researchers think this is to be expected while others do not.
> > >
> > > Although this may seem expected, we are not aware of previous work that is able to establish this limitation, especially theoretically and empirically. In fact, more embedding research seems to be trying to reduce the embedding dimensionality for storage constraints (e.g. Matroyoska Learning, etc). Thus, we aim to fill this gap with our paper.
> > >
> > > > one should not read the absolute numbers to intrepet anything but rather compare for different embedding dimensions? If this is clear already in the manuscript, please highlight the sentence(s) if not would you be willing to highlight it.
> > >
> > > For relative improvement with dimension this is the main point of the proof (Section 3), the free embedding experiments (Section 4), and in Section 5 empirical results we emphasize this in the Figure 1 caption and in the 2nd results paragraph starting on line 406.
> > >
> > > However, we do also think the absolute numbers are interesting, even if not the main contribution, which is why we discuss them also. As you mention, the format of the task (exact lookup) is more difficult for embedding models than lexical models but overall it is still not a very difficult task. A human (or even grep!) could solve this near instantly. We also believe this is still a useful skill for embedding models in many practical settings: e.g. for code functions/variable name lookup similar to Cursor/Claude Code, shopping queries where the exact brand matters, searching for entity’s names where exact lookup is important, etc.
> > >
> > > Given the constraints on needing queries to be able to return two unique documents, we are unable to fully disentangle the difficulty from representing all combinations and any bias from the way we instantiate the dataset in natural language. No matter how we instantiate it, the words we use will incur some bias. This is why we also include the free embedding experiments which show the difficulties models face in the ideal optimization case.
> > >
> > > Thus we do discuss both relative scores and absolute scores in the paper, but the relative scores of the dimensionality are the main contribution aligned with the proof and free embeddings.
> > >
> > > > if not would you be willing to highlight it
> > >
> > > **Thanks to your feedback we agree that this should be better highlighted.** We will add a paragraph in the results section to emphasize that these results are biased towards lexical models (and give the results of the synonym version to show that) and describe that dense models are still immensely useful for many tasks, but will not be able to completely solve the tasks where you need to represent all combinations (as the proof and free embeddings say also, better tying them together).
> > >
> > > We certainly don’t want to give people the wrong impression, so thank you for helping us see that that this should be better communicated!

---

> ### Comment · Reviewer_8eB8 · 2025-11-22
>
> I agree with you that I am not aware of any previous work which establishes this limitation around embedding dimension.
>
> ```Thanks to your feedback we agree that this should be better highlighted. We will add a paragraph in the results section to emphasize that these results are biased towards lexical models (and give the results of the synonym version to show that) and describe that dense models are still immensely useful for many tasks, but will not be able to completely solve the tasks where you need to represent all combinations (as the proof and free embeddings say also, better tying them together).```
>
> - thank you for agreeing to this.
>
>
> Thank you for the interesting conversation and engagement. I have updated my scores as I am happy with the clarifications. It perhaps is worth a thought for later experiments by you or the community as to how to experiment with this from a setup more geared towards sematic understanding to see if such embedding limits can be experimentally observed.

---

### Official Review · Reviewer_n8vt · 2025-11-01

**Soundness:** 4
**Presentation:** 3
**Contribution:** 3
**Rating:** 8
**Confidence:** 4

**Summary:**

This paper presents a significant theoretical and empirical challenge to the standard single-vector embedding paradigm for information retrieval. The authors argue that as embedding models are tasked with increasingly complex, instruction-based retrieval scenarios, they are running into a fundamental theoretical limit, not just a practical one that can be solved by larger models or more data. The core theoretical contribution is to formally connect the representational capacity of an embedding model to the sign-rank of the query-document relevance (qrel) matrix. The authors prove that for any fixed embedding dimension d, there is a hard upper bound on the number of unique top-k relevant document sets a model can represent, meaning some combinations of relevant documents are impossible to retrieve, regardless of the query.

To validate this theory, the paper introduces two key empirical results. First, it demonstrates using a "free embedding" optimization (a best-case scenario without natural language constraints) that this dimensional limitation is real and predictable. Second, it introduces LIMIT, a new benchmark dataset designed to be a simple, natural language instantiation of a task with a high-sign-rank qrel matrix. Experiments show that even state-of-the-art SOTA embedding models fail catastrophically on LIMIT. The work concludes that the community must reconsider the fundamental limitations of the current dense retrieval paradigm for future instruction-following tasks

**Strengths:**

1. The primary strength of this paper is its formal theoretical grounding combined with a rigorous empirical validation. Moving the discussion from a vague sense of "difficulty" to a concrete mathematical limitation (by connecting embedding capacity to the sign-rank of the qrel matrix) is a novel and important contribution for the field.
2. The LIMIT dataset is a significant contribution. It is not just another retrieval benchmark but a cleverly designed probe specifically built to empirically validate the paper's theoretical claims.
3. The paper is well-written, clear, and addresses a fundamental assumption in the IR/NLP community.

**Weaknesses:**

1. While the simplicity of the LIMIT dataset is a strength for the theoretical argument, it is also a potential weakness. The synthetic nature of the task ("who likes X?") and the explicit construction of an all-combinations qrel matrix make it unclear how often such high-sign-rank relevance structures appear in organic, real-world benchmarks (e.g., MS MARCO, BEIR). The paper claims instruction-following will create these, but doesn't demonstrate that existing complex benchmarks already suffer from this problem.
2. The theoretical argument is based on the inability to represent all possible $\binom{n}{k}$ top-k combinations. In practice, a model only needs to represent the subset of combinations that can be expressed by plausible natural language queries. This set is likely much smaller than the theoretical maximum. The paper doesn't fully bridge the gap between "all possible" and "all plausible" combinations.

**Questions:**

N/A

---

> ### Author Response · Authors · 2025-11-20
>
> Thank you for recognizing our work provides a “formal theoretical grounding combined with a rigorous empirical validation” and is “well-written, clear, and addresses a fundamental assumption in the IR/NLP community”!
>
> > The synthetic nature of the task ("who likes X?") and the explicit construction of an all-combinations qrel matrix make it unclear how often such high-sign-rank relevance structures appear in organic, real-world benchmarks (e.g., MS MARCO, BEIR).
>
> We agree that existing benchmarks (MS MARCO/BEIR) do not have this property, which is why we had to create the LIMIT dataset. However, recent benchmarks such as FollowIR, BRIGHT, and QUEST all show steps moving in this direction which, taken together, seems to indicate that the IR community is moving towards this “answer any instruction and relevance definition” setting.
>
> >  The paper doesn't fully bridge the gap between "all possible" and "all plausible" combinations.
>
> We agree and mention this in the limitations section in our paper (Appendix D). We will move this up to be more prominent. However, it is hard to know apriori which queries you are okay with your system failing as you cannot predict it beforehand. Thus, we advocate for new research that will move past this theoretical limitation.

---

### Official Review · Reviewer_i6hx · 2025-11-03

**Soundness:** 4
**Presentation:** 4
**Contribution:** 2
**Rating:** 4
**Confidence:** 5

**Summary:**

This paper provides fundamental limitation analysis of embedding models for IR, and shows that the such analysis results hold for many dataset instantiation.

**Strengths:**

The paper is well-written and easy to follow.
The experiments are well designed.
It is necessary to provide in-depth analysis for embedding models used for IR

**Weaknesses:**

See the below questions.

**Questions:**

The main goal of information retrieval is to improve the performance of a ranked list of documents in response to given a query. It is unclear how such theory analysis can help to boost the ranking performance. The motivations of finding the minimum embedding dimension are not convincing.

The row-wise order-preserving rank of A, as defined in Definition 1, is strictly constrained, where for **all** i, j, and k, they need to be subject to: if $A_{ij} > A_{ik}$ then $B_{ij}>$B_{ik}$. Can we relax the constraint? Let’s say: such that for xx present of i, j, k, if $A_{ij} > A_{ik}$ then $B_{ij}>$B_{ik}$?

In some IR task, the relevance scores can be set to be, e.g., 0 (non-relevant), 1 (relevant), 2 (highly relevant), i.e., using graded scores rather than binary scores to denote the relevance. Accordingly, in Definition 2, can we set A to be $A\in \{0, 1, 2\}^{m\times n}$ for instance?

 Are there any other limitations that the authors can provide theory analysis when applying embedding space for neural IR techniques?

What are the main differences between previous work that works on embedding dimension compression for embeddings used in IR tasks and the analysis the authors provide here?

Some writing/descriptions are not self-included and confusing. E.g., in the introduction section, the authors claim “Since embedding models use vector representations in geometric space, there exists well-studies fields of mathematical research (Papadimitriou & Sipser, 1982) that xx x”. The concept “mathematical research” is not self-included in this sentence.

Is the dataset publicly available for other researchers used in the experiments?

---

> ### Author Response · Authors · 2025-11-20
>
> Thank you for noticing that our work is “well-written” and “well designed” with “in-depth analysis”!
>
> > It is unclear how such theory analysis can help to boost the ranking performance. The motivations of finding the minimum embedding dimension are not convincing.
>
> Our work aims to show the limitations of these models, which will lead others to correct it, thereby increasing ranking performance in the future. As you mention, our work does not directly improve existing methods' scores, although we do show possible alternatives for improvement (e.g. multi-vector). **This is because our work only aims to show a limitation of single-vector models that needs to be fixed in the existing methods**, e.g. our work is purely an analysis/dataset paper contribution. However, our work shows that models can improve by increasing the dimensionality or using alternative architectures (multi-vector, hyperencoders, etc.), their performance will improve (e.g. they can get higher scores on LIMIT and similar datasets).
>
> > In some IR task, the relevance scores can be set … [to use] graded scores
>
> Great question! We agree that graded scores exist for many datasets – we simply chose to use binary scores for simplicity. However, forcing the model to understand graded relevance makes the problem even harder for models to learn, exacerbating the problems we illustrate. Your intuition is right and we will update the paper to include this discussion.
>
> > Are there any other limitations that the authors can provide theory analysis when applying embedding space for neural IR techniques?
>
> There are many ways to apply theory to vector embeddings (see Section 2.3 for a longer list). However, we focus on just this one aspect (dimensionality) in our paper, as it already required much analysis and experiments. We leave new areas of theoretical analysis to future work.
>
> > What are the main differences between previous work that works on embedding dimension compression for embeddings used in IR tasks and the analysis the authors provide here?
>
> If you mean works that aim to make the embedding dimension smaller (e.g. Matroskya Learning) then this is typically done for practical reasons and not to examine the theoretical ramifications. Our work implies that these compressed vectors might be limited due to the theoretical limitations that we present. **Thus, our works are complimentary.**
>
> >  The concept “mathematical research” is not self-included in this sentence.
>
> Due to space we simply used the reference to explain it. The reference is to the field of [communication complexity](https://en.wikipedia.org/wiki/Communication_complexity), a field of mathematical research which we lean upon and cite heavily in Section 2.3. However, we will expand this reference to be more clear what field of mathematical research we are referring to, thank you for the suggestion.
>
> > Is the dataset publicly available for other researchers used in the experiments?
>
> Yes, the dataset is publicly available (unanonymized). If you'd like to look at it, an anonymized version is at https://anonymous.4open.science/r/limit-C689/README.md.

---

### Meta-Review · Area_Chair_L1xw · 2026-01-05

**Summary:**

This paper investigates a fundamental theoretical limitation of single-vector embedding models for retrieval, proving that the number of distinct top-k result sets they can represent is bounded by the embedding dimension. The authors support this analysis with "free embedding" optimization experiments and introduce the LIMIT dataset, a synthetic benchmark designed to stress-test models under these constraints. Empirical results show that even state-of-the-art embedding models struggle on LIMIT, while sparse methods like BM25 excel on the lexical version but also fail when lexical cues are removed.

**Reviewer Concerns:**

Reviewer 8eB8's main concern about dataset/task mismatch was directly addressed. The authors created a synonym-based version of LIMIT, demonstrating that BM25's performance plummets without exact lexical overlap, confirming that the issue is not merely about favoring sparse methods. The reviewer found this explanation satisfactory and raised their score.

Reviewer n8vt's concern about the real-world prevalence of high sign-rank structures was acknowledged. The authors argued that emerging instruction-following retrieval benchmarks (e.g., FollowIR, BRIGHT) are moving in this direction, justifying the investigation.

Clarity issues (raised by 8eB8 and i6hx) regarding missing figures, variable definitions, and sentence structure were committed to be fixed.

**Reviewer Scores:**

Reviewer i6hx (Score: 4 -> ?): The reviewer did not participate post-rebuttal. Their concerns (clarity, motivation) were addressed textually by the authors. It is plausible, though not certain, that a fuller discussion could have led to a score increase.

Reviewer n8vt (Score: 8 -> 8): This reviewer's positive assessment was consistent throughout. Their constructive criticism about real-world relevance did not diminish their strong support for acceptance.

Reviewer 8eB8 (Score: 4 -> Positive): Through constructive dialogue and additional experiments, the authors resolved this reviewer's primary objection. The reviewer explicitly stated they were "happy with the clarifications" and updated their scores positively, likely to a 5 or higher.

---

### Decision · Program_Chairs · 2026-01-26

Accept (Poster)